



Reversing land degradation through grasses: a systematic meta-analysis in the Indian tropics
**Debashis Mandal[1], Pankaj Srivastava[1], Nishita Giri[1], Rajesh Kaushal[1], Artemi Cerda[2],**
**Nurnabi Mehrul Alam[1]**
[1]ICAR-Indian Institute of Soil and Water Conservation
218, Kaulagarh Road, Dehradun-248195, Uttarakhand, India
[2]Department of Geography, University of Valencia, Blasco Ibañez, Valencia, Spain
Corresponding Author: Dr. D.Mandal, ICAR-National Fellow, Soil Science and Agronomy
Division, 218, Kaulagarh Road, Dehradun-248195, Uttarakhand, India
Email: dmandalcswcrti@gmail.com, pksnbri@gmail.com
**Abstract**





The present study critically analyzes the effect of grasses in reversing the process of land
degradation using a systematic review. The collected information was segregated under three
different land use and land management situations. Meta-analysis was applied to test the
hypothesis that use of grasses reduce runoff and soil erosion. Effect of grasses was deduced
for grass strip and in combination with physical structures.  Similarly, the effects of grasses
were analyzed in degraded pasture lands. The overall result of the meta-analysis showed that
infiltration capacity increased approximately two-fold after planting grasses across the slopes
in agricultural fields. Grazing land management through *cut and carry* system increased
conservation efficiencies by 42% and 63% with respect to reduction in runoff and erosion,
respectively. Considering comprehensive performance Index (CPI) it has been observed that
hybrid napier (*Pennisetum purpureum*) and sambuta *(Saccharum munja)* seem to posses the
best desirable attributes as effective grass barrier for western Himalaya and eastern Gahts
while natural grass *(Dicanthium annulatum) and broom grass (Thysanolaena maxima)* are
found to be most promising grass species for Konkan region of western Ghat and north
eastern Himalayan region, respectively. In addition to these benefits, it was also observed that
soil carbon loss can be reduced by 83% with the use of grasses. Overall, efficacy for erosion
control of various grasses was more than 60% hence their selection should be based on the
production potential of these grasses under given edaphic and agro-ecological condition.
**Key-words**
**Contour grass barrier, Conservation efficiency, Grazing, Reverse land degradation, Soil**
**erosion,**


**1 Introduction**



Water erosion is the main cause of land degradation affecting about 2 billion ha area
throughout the world with a largest part in tropics which affect two most important natural
resources, namely, soil and water (Mandal and Sharda, 2011a; De Oliveria et al., 2010;
Keesstra et al., 2014; Novara et al., 2011; Seutloali and Beckedahl, 2015; Novara et al.,
2016). Worldwide loss of water and sediment due to soil erosion is a major environmental
threat (Prosdocimi et al., 2016; Pimentel, 1993). Soil erosion is accelerated due to high
rainfall intensities (Keesstra *et al.*, 2016), steep slopes (Beskow et al., 2009) and fragile
nature of top soil (Lal, 1998; Rodrigo-Comino et al., 2016; Ochoa et al., 2016 ). Many parts
of the tropics in India have high annual rainfall confined to only four to five months (June-
September). During the seven to eight months dry period, scarcity of water causes a severe
shortage of fodder in farmlands which leads to increase grazing pressure on forest and
community lands. Nearly, a third of the fodder requirement in India is met from the forest
resources in the form of grazing and cut fodder (MoEF, 1999). The process of land
degradation in croplands and grasslands has been accelerated mainly by inappropriate land
use (Nearing et al., 2005; Mandal et al., 2010) and mismanagement (Kagabo et al., 2013).

Generally, conservation planning needs the soil loss tolerance value which is

considered as the higher limit of soil erosion rate that can be allowed without long term land
degradation (Jha et al. 2009). Strategies to reverse land degradation are critical since soil is a
non-renewable resource (Mandal and Sharda, 2011b; Mandal et al., 2010). Soil erosion rates
more than tolerance values are considered non acceptable (Mandal and Sharda, 2013) which
leads to irreversible land degradation and need to be reduced through appropriate soil
conservation measures (SCM) (Biswas et al., 2015) The physical structures to check soil
erosion are proven effective but are cost intensive. Biological methods of soil and water
conservation, especially, grass based methods have been reported to be very cost effective
and suitable for sloppy lands. Perennial grasses provide ground cover throughout the year and



helps in reducing runoff and soil loss when used as barriers along the contour particularly in
hill slopes (Dhruvanarayana and Ram babu, 1983). Grasses are the key component in many
ecosystems of the world (Parras-Alcantara et al., 2015; Hu et al., 2016; Mekonnen et al.,

2016).

Grass species, in particular, have tremendous potentialities in soil conservation as
grass roots have a great binding influence on soil particles (Ovara et al., 2013; Ola et al.,
2015). Due to resource scarcity and multiple competing enterprises that characterise most
farming situations of rural India, farmers often lack adequate resource to invest in physical
soil conservation structures. Thus, the usefulness of grasses as vegetative barrier is an
alternative to the physical soil structures. Basically these contour vegetative barriers/grass
filter strips help in reducing soil erosion by acting as porous barriers which subsequently slow
down the flow of runoff (Angima et al., 2001; Mutegi et al., 2006).
The hilly region of India is characterized by geological fragility, land marginality and
vulnerability (Mandal and Sharda, 2013). The croplands in sloppy areas suffer from excessive
soil erosion and erosion induced nutrient depletion. Soil erosion in these areas ranges
between 20-40 Mg $ha^{-1}$ $yr^{-1}$ as compared to the national average of 16.35 Mg $ha^{-1}$ $yr^{-1}$
(Dhruvanarayana and Ram babu, 1983). Such high rates of soil erosion result in considerable
depletion of nutrients from the top soil which in turn causes poor productivity of crops.
Research evidence from the land subjected to shifting cultivation reported that about 600 Mt
(million tons) of soil is eroded annually which led to losses of 258,000, 73,000 and 179,000
tonnes of N, $P_2O_5$ and $K_2O$, respectively (Kumar, 2011). Soil erosion has been pointed as one
of the important reason for the land abandonment by many farmers in sub-tropical hilly areas
of India (Rao and Pant, 2001).



The grasslands in middle and lower Himalayas are generally in the most neglected
state with low productivity. In this predominantly grazing region, excessive reliance on
animal husbandry under a growing population has exerted great pressure on the land. In
tropical India, an average of 42 animals graze on a hectare of land compared to maximum
threshold level of 5 (Sahay, 1999).   Raising and maintenance of perennial grasses on
degraded soils has been suggested as a means to improve soil quality and sequester carbon in
the soil. Several studies have shown that the inclusion of grasses in the agricultural landscape
often improves the productivity of system while providing opportunities to create carbon (C)
sinks (Ghosh et al., 2009; Cogle et al., 2011; Huang et al., 2010; Mutegi et al., 2008). Soils
typically account for 70-90% of the total carbon sequestered in a grassland ecosystem
(Batjes, 2001).

In India most of the studies on the role of grasses as vegetative/filter strips have been

done in isolation with fewer slope categories and with limited objectives restricting to soil
erosion (Njoroge and Rao, 1994). Similarly, the studies on grazing land management are also
very scarce. We present here an analysis on the potential of grasses for reversing land
degradations for which the meta-analysis was carried out. The objective of this study is to
determine the effect of grasses in arresting soil loss, runoff, moisture conservation and carbon
build up in soils. Based on such information, conclusion regarding reversing land degradation
through grasses can be drawn wherever similar land conditions are known.
**2 Material and methods**
Information on the usefulness of grasses in soil and water conservation was collected from
published literature (Table 1 a and 1b). Keeping in view the role of grasses for arresting soil
loss and runoff, all data were reoriented under three different categories viz; (i) role of
grasses as vegetative barrier, (ii) complementary role of grasses with physical soil structures



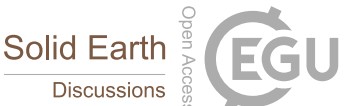

and (iii) management of grazing lands. A total of 83 studies comprising 19 different sites in
varied agro-climatic region were included in the data set for the analysis (Table 1a and1b).
Fifty four of these studies were related to contour grass barrier (CGB), 12 related to grazing
and 17 related to complementary role of grasses.

Meta-analysis was applied to test the hypotheses about role of grasses in reducing soil

erosion by combining data from several experiments. The technique has been extensively
used in natural resource management studies (Ilstect et al., 2007, Poeplau and Don, 2015;
Osenberg et al., 1999).

We aim to synthesise and discuss the fact that can be drawn from the past scientific

studies pertaining to the effect of grasses in arable and non-arable lands on one of the key
determining soil processes, namely reduction in soil and water losses and enhancement of
infiltration. We systematically used quality criteria to select studies to which we applied meta
analysis in order to produce a combined data set with the condition that a reference bare
land/fallow land had to be present with all the study sites. The reference sites were adjacent
to the grass treated filed/plots within the same landscape and similar slope. Therefore, we
excluded studies where the reference site was either missing or was away from the field
study. The conservation use efficiency (CUE) was calculated by the following formula
(Khola and Sastry, 2005).

(The water or soil runoff rate before the conservation measure) - (The water or soil
runoff after the conservation measure) X 100
CUE= ───────────────────────────────────────────────────────────
The water or soil runoff rate before the conservation measure


Data were analyzed using the SPSS (version 17). The Analysis of Variance (ANOVA)

was conducted to test the significant difference between different treatments. Initially, a t-test
was conducted to test whether the impact of two treatments (without grass and with grass)



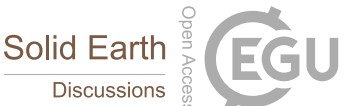

were significantly different. Protected least significant difference (LSD) at P=0.05 was used
to separate the means for all the three different categories of data (Fisher, 1935). A separate t-
test was also used for different slope classes to evaluate the performance of CGBs on the
reduction of soil and water loss and enhancing crop yield.
Relative performance of different grasses used as CGB was evaluated by using a
comprehensive performance index (CPI). The following formula was used to compute CPI
values of different grasses (Sudhishri et al., 2008).

$$CPI = \sum_{i=1}^{n} W_i R_i$$



Where CPI is comprehensive performance index of the grass species, $W_i$ is weightage of the
$i$th parameter, $R_i$ is rating (scoring) of the $i$th parameter based on its observed value. A total
of six attributes namely, infiltration rate, soil loss, root binding capacity, maximum sod
forming depth, fodder/commercial value and cost of establishment (Table 2) were used for
computing CPI.
Additionally, relative reversibility of erosion/water loss and relative yield gained due to
adoption of CGBs were computed by using the following formulas, respectively.
**Relative reversibility of erosion/water loss:**
Δ Erosion/ Δ Runoff = $\dfrac{\text{Erosion/ Water loss without CGB} - \text{Erosion/ Water loss with CGB}}{\text{Mean erosion/ Water loss}}$ X100







**Relative yield gain:**
$\Delta$ Yield gain = $\dfrac{\text{Mean yield with CGB} - \text{Mean yield without CGB}}{\text{Mean Yield}}$ X 100
**3 Results and discussion**
**3.1 Contour Grass Barrier (CGB)**
India is the home of about 1225 species of grasses, majority of which grows well in tropical
and subtropical region (Prakash et al., 1999). These grasses can be used as live bunds in
arresting soil erosion. Efficacy of CGBs in increasing the opportunity time for infiltration and
consequent profile recharge was also reported by other researchers (Sharma et al., 1997;
Prakash et al., 1999). In this meta-analysis, based on 25 observations, we quantified the
general potential of vegetative barriers to reduce run off and soil loss (Table 3). The overall
result of the meta-analysis showed that infiltration capacity increased approximately two-fold
after planting grasses across the slopes in agricultural fields (95% confidence level).
However, it is interesting to note that the mean runoff values were statistically insignificant in
case of combined treatment of grasses along with structural measures. This may be due to
very high standard deviation (SD) values obtained for vegetative barrier. These higher values
indicate lot of heterogeneity in the observation which needs to be verified. Although 70%
data showed similar variation, however, few higher values were not in expected lines which
might have caused this uncertainity. In case of Doon valley region, comparing the impacts on
soil wetting pattern, infiltration rate and sorptivity, it was observed that *Chrysopogon fulvus*
was most promising grass species. However, in Doon valley region *Panicum maximum* is
identified as most effective grass barrier with maize, but more research is required with
*Chrysopogon fulvus* because the rooting pattern, soil wetting, infiltration rate and other
properties of this grass shows great potentiality to be used as contour grass barrier in valley as



well as in hilly areas (Mandal and Jayaprakash, 2009). It was identified that *Saccharum*
*munja* and *Eulaliopsis binata* are two most effective grasses for *Shivalik* region of Punjab and
Haryana while hybrid napier and *Panicum maximum* are very effective in humid tropical
regions of lower Himalaya.

Run off and soil loss values in CGB plots were lower than the control plots. The data

show that run off varies between 11.26% and 62.60% with the mean value 37.71% and soil
loss varies between 0.53 Mg ha$^{-1}$ yr$^{-1}$ and 30.90 Mg ha$^{-1}$ yr$^{-1}$ with the mean value 9.56 Mg
ha$^{-1}$ yr$^{-1}$ in control treatments (Table 3). With CGB, the runoff data varies between 5.87% and
44.10% with the mean value 20.93% and soil loss varies between 0.50 and 18.70 Mg ha$^{-1}$ yr$^{-1}$
with the mean value 3.93 Mg ha$^{-1}$ yr$^{-1}$. The study revealed that on an average the overland
flow reduced by 45% compared to control. CGB facilitated the appearance of backed-up
water above the filter strips, which resulted in sedimentation and substantial reduction in soil
loss. The analysis of the data indicated that as the rain proceeded, overland flow moved down
slope into the grass hedges and water backed-up behind them, giving more opportunity time
for the water to infiltrate the soil. Experiments conducted by Becker (2001) reported reduced
soil erosion by parallel strips of stiff-stemmed grass planted along the contour lines. Over and
above, the amount of transported soil reduced by 59% in case of grass barriers than that of the
control. A substantial reduction in runoff from 37.71% in control to 20.93% in CGB was
observed. Vegetative barriers reduced the soil loss from 9.0 Mg ha$^{-1}$ yr$^{-1}$ to 3.0 Mg ha$^{-1}$ yr$^{-1}$.
The CUE of vegetative barrier was found to be 44.56 and 59.04%, for runoff and soil loss,
respectively. These findings are in conformity with the results reported by Gilley et al. (2000)
who have summarized that grass hedges have the potential to reduce runoff by 52% and soil
loss by 53% under no-till conditions. Globally, most researchers in tropical region have used
vetiver grass (*Vetiveria zizanioides*), eastern gamagrass (*Tripsacum dactyloides*) due to their
special characteristics with stiff, erect and coarse stems (Rachman et al., 2004, 2005;



Janushaj, 2005). Such species are perennial in nature thus show a good protective cover
throughout the year in warm humid topics.
In terms of soil loss, the vegetative barrier of *Panicum maximum* showed promising
performance with average rate of soil loss between 2.74 Mg ha$^{-1}$ yr$^{-1}$ and 7.93 Mg ha$^{-1}$ yr$^{-1}$ in
north western Himalayan region which indicated that soil loss can be effectively brought
below tolerance limit by adopting such SCM (Mandal et al., 2006). Considering the
advantages of contour grass strips, over the mechanical measures, due to their less cost and
minimum removal of the fertile top soil many organizations are promoting this practice as an
effective measure to reduce erosion (ASAE, 1981; Hudson, 1981; Mulugeta, 1988;
Turkelboom et al., 1994). Moreover, CGB is comparatively simple and easy to establish
(Grunder, 1988), while mechanical measures are too expensive, are difficult to maintain in
the long run (Rodriguez, 1997) and are time consuming (Tripathi and Singh, 1993).
Additional advantages with regard to establishment and stabilization of the grass strip is that
it needs very less attention to form a terrace while mechanical measures need regular
maintenance to keep their effectiveness (Welle et al., 2006).
A study revealed that *Panicum maximum* provided 56% of coverage after three years
of planting. The coverage increased progressively from 23% in 1st year to 56% in 3rd year.
Similarly, Vetivercoverage increased from 29% in 1st year to 75% in 3rd year (Shrimali,
2000). Vetiver grass distinctively showed highest reduction in annual runoff and soil. This
was attributed to the fact that the erect and rather stiff leaves and stems of vetiver grass
retarded more runoff flow and acted as filter to more sediment. Similar performance level of
vegetative barrier was also reported by Rao et al. (1991) and Laing and Rupenthal (1991).
This is also in conformity with the results of Patil et al. (1995b), who recorded 41.4% lesser
runoff for vetiver over control. Similar results had been obtained by Tangtumniyom *et al*.
(1996) for a cassava crop on a 5% slope where vetiver was used as vegetative barrier. The




241 effect produced by *Cenchrus ciliaris* planted at 10-m spacing was also comparable to that of

242 vetiver at 10 m, which recorded a mean annual soil loss of 3.39 Mg ha$^{-1}$ (Jagannathan et al.,

243 2000).

244  The conservation of soil and water in CGB varies with grass types and site conditions

245 in different regions. However, *Pennisetum purpureu*m, *Panicum maximum* and *Eulaliopsis*

246 *binata* were very effective for lower Himalayan and Shiwalik region. Results from different

247 studies across the country showed that due to the large amount of green phytomass, profuse

248 tillering and dense rhizomatous network of roots, runoff and soil losses were significantly

249 reduced with barrier of *Pennisetum purpureum.* For different regions of India including

250 Andhra Pradseh, Haryana, Karnataka, Madhya Pradseh, Maharashtra, Orissa, Punjab,

251 Tamilnadu and Uttarakhand suitable grasses for CGB are given in Table 4. In situation where

252 fodder requirements are high *Pennisetum purpureum* mounted as a barrier would be

253 beneficial, while in those areas where soil conservation is utmost important, *Eulaliopsiss*

254 *binata* or aromatic grasses such as palmarosa (*Cymbopogon martinii*)  or vetiver (*Vetiver*

255 *zizanoides*) grass would be reasonable choice.

256  Analysis of variance through t-test of soil loss, run off and yields of crops indicated

257 that loss of water was significantly less in CGB treated sites in <2% slopes (Table 5). The

258 water loss provided by CGBs compared to control was 16% Vs 27% for < 2% slope.

259 However, the similar trend was not observed in 2-4% slope range. Interestingly the soil loss

260 was significantly less in CGB treated sites in higher slopes (2-4 and >4% slopes).

261 Variations in soil erosion amounts paralleled to some extent to those of runoff in all the slope

262 classes except in lower slope range (Table 5). The protective action of various CGBs are very

263 clearly shown by the soil loss values which reflect that between 141% and 107% reversibility

264 in soil loss can be achieved through adoption of CGB. The relative reversibility of water loss



provided by CGBs compared to control was 52.6% and 55.5% for < 2% and >4% slopes,
respectively. Favourable soil condition created by CGBs resulted an increase in yield in all
slope ranges. The significantly higher yield in CGB treated sites have resulted from either
better moisture regime or higher nutrients or by both depending on the detention of runoff
and deposition of fertile sediment by the CGBs. The relative yield gained by CGBs varied
between 44% and 53% with highest value in 2-4% slope.
Clear picture about the relative merit of CGBs was determined through development
of CPI for different grasses (Table 6). Hybrid Napier *(Pennisetum purpureum)* seems to
posses the best desirable attributes for soil and water conservation with highest CPI value of
0.81. On the other hand *Saccharum munja* had fairly good merit (0.79) in conserving soil
and water and has both fodder and commercial values. Similarly *Dicanthium annulatum* with
CPI value of 0.77 has an edge over broom grass (0.72). However from farmer adaptation
point of view, both *Saccharum munja* (0.79) and *Thysanolaena maxim*a (0.72) grass are most
preferred species especially in shifting cultivation area of Eastern Ghats and north eastern
hilly region of India.
**3.2 Complementary role of grasses with physical soil structures**
Grasses, shrubs and tree barriers in combination with structural measures (bioengineering
measures) are known to be beneficial for soil and water conservation and have many relative
advantages over structural interventions. Reinforcement by live roots which bind soil
particles and underground decomposed biomass provides stability to aggregated soil. Plant
detritus on the soil surface act as a cushion for dissipating kinetic energy of rain drops. This
above ground biomass upon its subsequent decomposition also adds to the soil humus and
increases infiltration, soil water holding capacity as well as stability of aggregates (Prakash *et*
*al.*, 1999).



The data from Table 3 show that the use of grasses led to significant decrease in
runoff from 25.53% in control to 9.37% with structural conservation measures. Soil loss also
has significant decrease from 1.88 Mg ha$^{-1}$ yr$^{-1}$ in control to 0.73 Mg ha$^{-1}$ yr$^{-1}$ in structural
conservation measures (Table 2). The run off varies between 17.00%  and 48.50% with the
mean value of  25.53% and soil loss varies between 1.53 Mg ha$^{-1}$ yr$^{-1}$ and 3.26 Mg ha$^{-1}$ yr$^{-1}$
with the mean value of 1.88 Mg ha$^{-1}$ yr$^{-1}$ in control. The runoff varies between 0.40% and
15.30% with the mean value of 9.37% in combined treatment (grass along with structural
measures). Similarly, analysis of the data revealed that the impact of grasses was more
pronounced along with soil and water conservation measures in minimizing the losses of soil
and water.  Over and above, the complimentary action shows water saving by 63% and soil
saving by 61%.
Earthen bund and earthen bund with broom was found to be more effective in soil
moisture conservation at 4% and 8% slope as compare to other treatments (Figure 1). In
comparative study conducted on *Pennisetum* and *Arundinella* barriers in combination with
soil conservation measures, a substantial reduction (65-88% and 15-38%, respectively) in
overland flow compared to the control plots had been reported (Huong et al., 2010).
**3.3 Management of grazing lands**
In India about 12.0 m ha of area is represented by permanent pasture and grasslands, majority
of which is confined to the tropical areas (Roy and Singh, 2013). Since this pastureland and
grasslands are severely affected by soil erosion, special attention should be given to their
management to reverse the process of degradation. Our synthesis of the meta analysis
revealed that by managing the grassland with cut & carry system, rotational grazing and
control grazing can greatly reduce the water and soil loss and helps in the reversing the land
degradation process. Similar phenomena have been reported by Misri (2003) and Pathak and



Dagar (2015), especially, for the lower Himalayan and Shivalik grassland where severe biotic
pressure is imposed by both sedentary and migratory graziers. The grazing intensity in the
country is as high as 12.6 adult cattle units per hectare (ACU ha$^{-1}$) as against the carrying
capacity 0.8 ACU ha$^{-1}$ (GOI, 2015).
The data (Table 3) show that run off varies between 11.30% and 33.40% with the mean value 24.33%
and soil loss varies between 1.52 Mg ha$^{-1}$ yr$^{-1}$ and 3.28 Mg ha$^{-1}$ yr$^{-1}$ with the mean value 2.58 Mg ha$^{-1}$
yr$^{-1}$ in control plots  (without grazing management). The management of grazing lands (cut and carry
system, rotational grazing and control grazing) significantly reduced the runoff ranging between
6.60% and 22.20% (with the mean value 14.12%) and soil loss ranging between 0.58 Mg ha$^{-1}$ yr$^{-1}$ and
1.30 Mg ha$^{-1}$ yr$^{-1}$ (with the mean value 0.95 Mg ha$^{-1}$ yr$^{-1}$). A total of 12 studies on grazing land
management revealed that the benefits of stall feeding and controlled grazing could save about 42%
water loss and 63% soil loss in sloppy lands.  The mean runoff in grazing management practices was
significantly reduced from 24.33% to 14.12%.  This may be due to higher green cover and biomas
production under improved management. Grazing land management of *Chrysopogon fulvus,*
*Heteropogon contortus* and *Panicum maximum* have shown potentiality to produce 40 t ha$^{-1}$, 8.5 t ha$^{-1}$
and 110 t ha$^{-1}$ green biomass yields, respectively (Rana, 1998; ICAR, 2006; Ghosh et al., 2009;
Pathak and Dagar, 2015). The average soil loss was significantly reduced from 2.0 Mg ha$^{-1}$ yr$^{-1}$ to 0.95
Mg ha$^{-1}$ yr$^{-1}$ by the imposition of grazing and grassland practices. However, some researchers
demonstrated that the grass steppe is more resistant to land degradation than shrub steppes (Palacio et
al., 2014 and they contribute to increase the biodiversity and to improve the soil quality (Costa et al.,
2105; Gao-Lin et al., 2016)

*Dicanthium annulatum* cover was found to reduce the runoff and soil loss by 35.45-

51.40% and 71.90-81.08%, respectively, in slightly to severely degraded lands in lateritic soil
of Konkan regions in India (Figure 2 and 3).  In this region *Dicanthium annulatum* yielded
about 25-30 t ha$^{-1}$ of green biomass under improved management. The investigation further
suggested that carbon loss can be reduced to the extent of 88.36 - 83.12 % in slightly and



severely degraded lands in the same region (Figure 4). The study also indicated that carbon
sequestration rate up to 100 kg ha $^{-1}$ yr$^{-1}$ can be achieved by the use of grass strips running
across the slope especially in laterite soils of *Konkan* region (Kale et al., 1993). About 6 fold
increase of SOC content in soil has been observed in barren lands of *Shivalik* region through
rehabilitation by *Arundo donex*. Grazing management typically leads to a 3% annual increase
in soil carbon (Conant et al., 2001). Duran and Rodriguez (2008) highlighted that grasses
provide perennial protection and minimal erosion as they provide complete ground cover
(Brindle, 2003). In Mediterranean region, based on 20 paired plots study, Keesstra et al.
(2016) reported that runoff sediment concentration was 45.5 times higher in cleaned
cultivation plots compared to covered plots. They further reported that erosion rate was below
the soil loss tolerance limits under surface covered conditions. It is noticeable that the loss of
vegetation cover leads to increase surface instability and poor regeneration which in turn set
a vicious cycle in motion.
In the hilly region of north-eastern Himalaya, the alternative land use systems help in
reducing soil erosion systems and SOC loss to a substantial extent. Higher root-biomass of
the grasses, particularly *Paspalum, Congosignal, Hamil* and *Makunia* due to greater water
transmission resulted in higher SOC in the soil profile. Following addition of organic matter
through continuous root decay of these grasses, water holding capacity of the soil increased
as a result of the increased specific surface area. Additionally, these grasses helped in
improving soil quality including soil hydro-physical characteristics and biological activities.
Such improvement in soil properties have a direct bearing on C-sequestration (5 fold increase
in SOC over control), long-term sustainability, reducing soil erosion (2-3 fold increase in
structural stability over control) in a complex, risk prone fragile ecosystem (Ghosh et al.,

2009 )




## 4 Conclusions

Human induced changes due to land use intensification and overgrazing caused some severe and extreme state of land degradation that may prove to be more difficult to restore under the ongoing practices. The present meta-analysis clearly revealed that suitable conservation measures especially, the vegetative and biological practices greatly assist in reversing the land degradation process for both cropland and grasslands.

Most soil erosion control measures implemented on cultivated fields are physical structures. However, these physical structures were reported to be less acceptable due to high cost of their construction and maintenance. The Meta analysis clearly showed that grass barriers potentially reduce runoff and soil loss by up to 86.8% and 97.32 %, respectively. The relative yield gained of various crops through CGBs at different slopes varied between 44% and 53%. However, the effectiveness of grass barrier, as reported by several studies, is site-specific and depends mostly on slope gradient, runoff volume and flow rate, size and density of sediment particles, grass species, density, interval and width of grass strips, underlying soil properties, and rainfall intensity and duration. According to farmer's criteria based on comprehensive performance index, the study revealed that *Pennisetum purpureum* was most preferred grasses followed by *Saccharum munja* and *Dicanthium annulatum*. Considering the CPI values it is apparent that *Saccharum munja* (Sambuta) and *Thysanolaena maxim*a (hill broom) are two important bio-remediation options for reclamation of shifting cultivation of north eastern hill region and eastern Ghat of India.

The present analysis also indicated that grass must be used as vegetative strip to maintain soil quality in sloppy arable areas (8.5 m ha) of Indian hilly regions. Special emphasis on establishing grasses should be given to about 3 m ha degraded pasture lands and 3.5 m ha shifting cultivation areas in India to reverse the land degradation. Overall, we



conclude that the use of grass barriers alone or in combination with structural measures and

grassland management were  effective and efficient for decreasing soil and water loss on

sloppy  croplands in tropical and sub-tropical regions of India. Thus, these practices should

be intensively recommended and used widely in similar climatic regions. Similarly, the

reduction in grazing intensity needs to be advocated for about 12 m ha of permanent pasture

lands.

*Acknowledgement*

We owe our sincere thanks to the ICAR for funding this project (ICAR- National Fellow) and

Director, Indian Institute of Soil and Water Conservation (IISWC) for providing facilities and

support.

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



**Table 1a.** Details of the experiments and sources of data used in the study

| Vegetative barrier | With contour Grass barrier | | | Without Control Grass barrier | | Soil type and climate | Source |
|---|---|---|---|---|---|---|---|
| | Name of grass | Soil Loss (Mg ha$^{-1}$ yr$^{-1}$) | Runoff (%) | Soil Loss (Mg ha$^{-1}$ yr$^{-1}$) | Runoff (%) | | |
| | *Dichanthium annulatum* | 1.0 | - | 16.68 | - | Red soil (Rhodustalfs), Hot sub-humid | Lal et al., 2004 |
| | | 4.2 | 33 | 10.8 | 48 | | |
| | | 0.1 | 7.7 | 6.35 | 64.8 | Inceptisols, semi arid Red, hot sub-humid | Rao & Pande , 2014 Sharma, 1999 |
| | (*Tripsacum laxum*) | - | 19 | 19 | 29 | Red( laterite), Warm sub humid | Madhu et al. 2004 |
| | *Panicum maximum* | 2.47 | 20.7 | 8.1 | 40.9 | Alluvial, subtropical | Sharda et al., 2002 |
| | | 5.62 | 34.3 | 20.6 | 48.3 | Alluvial, subtropical | Ojasvi et al., 2000 |
| | | 7.54 | 28.6 | 30.9 | 37.9 | Alluvial, subtropical | Ojasvi et al., 2000 |
| | | 7.93 | 17.04 | 15.26 | 22.79 | Alluvial, Subtropical | Khola, 2000 |
| | Natural | 2.17 | 35.08 | 5.08 | 54.5 | Laterite, Hot-sub-humid | Yadav et al., 2000 |
| | | 0.5 | 22.7 | 1.05 | 49.6 | | Rao et al., 1998 |
| | | 1.37 | 39.9 | 2.16 | 54.8 | Laterite, Hot-sub-humid | Rao et al., 1998 |
| | | 1.02 | 44.1 | 1.72 | 59.1 | | Rao et al., 1998 |
| | | 0.59 | 5.87 | 3.12 | 12.08 | Laterite, Hot-sub-humid | Kale et al., 1993 |
| | | 0.76 | 10.2 | 4.4 | 16.95 | | Kale et al., 1993 |
| | | 1.36 | 13.36 | 4.84 | 20.1 | Laterite, Hot-sub-humid | Kale et al., 1993 |
| | *Cenchrus ciliaris* | 0.6 | 16.25 | 7.05 | 46 | Black (Inceptisol) Hot-semiarid | Nalatwadmath & Rao., 2000 |
| | | 0.81 | 21.9 | 1.39 | 29.5 | | Katiyar et al., 2007 |
| | | 0.5 | 6.6 | 16.08 | 68.7 | Red, hot sub-humid | Sharma, 1999 |
| | *Vetiveria zizanioides* | 9.02 | 19.17 | 15.26 | 22.79 | Alluvial, Subtropical | Khola, 2000 |
| | | 0.29 | 7.29 | 0.53 | 11.26 | Red, hot sub-humid | Katiyar et al., 2007 |
| | | 1.29 | 25.4 | 6.35 | 64.8 | Red, hot sub-humid | Sharma, 1999 |
| | | 0.5 | 8.6 | 0.7 | 20.7 | Red, hot sub-humid | Sharma & Bhatt, 1996 |
| | *Thysanolaena maxima* (Broom) | 15.7 | 14.2 | 19 | 17 | Red laterite (Alfisol) Hot sub-humid | Sahoo & Adhikari, 2014 |
| | | 18.7 | 17.3 | 23.9 | 23.5 | | Sahoo & Adhikari, 2014 |
| | *Heteropogon hamata* | 0.59 | 20.8 | 1.39 | 29.5 | Red laterite (Alfisoil) Hot sub-humid | Katiyar et al., 2007 |

| Grassland management | | | | | | | |
|---|---|---|---|---|---|---|---|
| **Grass** | Grass improvement | | | Traditional Grass | | Soil type and climate | Source |
| | Species | Soil loss (Mg ha$^{-1}$ yr$^{-1}$) | Runoff (%) | Soil Loss (Mg ha$^{-1}$ yr$^{-1}$) | Runoff (%) | | |
| | *Cynodon dactylon* | 0.06 | 35 | 3.28 | 54 | Red, hot sub-humid | Hazra & Singh, 1987 |
| | *Cenchrus* | 0.13 | 33 | 3.28 | 28.12 | Red, hot sub-humid | Hazra & Singh, 1987 |



| *ciliaris* | 2.14 | 16.8 | 3.33 | | Black, hot semi arid | Ilango et al., 2002 |
|---|---|---|---|---|---|---|
| *Panicum antidotale* | 0.43 | 36 | 3.28 | 54 | Red, hot sub-humid | Hazra & Singh. 1987 |
| *Pennisetum polystachyon* | 0.07 | 27 | 3.28 | 54 | Red, hot sub-humid | Hazra & Singh., 1987 |
| *Urochloa stolonifera* | 0.08 | 32 | 3.28 | 54 | Red, hot sub-humid | Hazra & Singh., 1987 |
| *Cymbopogon martini* | 1.08 | 11.32 | 3.33 | 28.12 | Black, hot semi arid | Ilango et al., 2002 |
| *Dicanthium annulatum* | 1.98 | 12.56 | 3.33 | 28.12 | Black, hot semi arid | Ilango et al., 2002 |
| *Vetiver zizanioides* | 2.61 | 18.4 | 3.33 | 28.12 | Black, hot semi arid | Ilango et al., 2002 |

| Grazing | Open Grazing | | Grazing Management | | Soil type and climate | Source |
|---|---|---|---|---|---|---|
| Treatment | Soil loss (Mg ha$^{-1}$ yr$^{-1}$) | Runoff (%) | Soil Loss (Mg ha$^{-1}$ yr$^{-1}$) | Runoff (%) | | |
| | 2.35 | 27 | 0.85 | 19 | Red, hot sub-humid | Hazra & Singh, 1986` |
| | 3.28 | 22 | 0.58 | 11 | Red, hot sub-humid | Hazra & Singh, 1986 |
| | - | 24 | - | 13.9 | Alluvial, Hot sub humid | Bhatt, 2013 |
| | - | 11.3 | - | 6.6 | | Bhatt, 2013 |
| | 1.52 | 21.6 | 1.52 | 10.2 | Alluvial, Hot sub humid | Rao & Reddy, 1996 |
| | 3.26 | 29.34 | 0.84 | 15.35 | | Rao & Reddy, 1996 |
| | | 20.40 | 1.18 | 9.6 | Black, Hot semi arid | Khola, 2004 |
| | | 33.40 | | 19.2 | Black, Hot semi arid | Khola, 2004 |
| | | 29.90 | | 22.2 | Black, Hot semi arid | Khola, 2004 |

Combination

| | With Grass (SWC) | | Without Grass (SWC) | | Soil type and climate | Source |
|---|---|---|---|---|---|---|
| | Soil Loss (Mg ha$^{-1}$ yr$^{-1}$) | Runoff (%) | Soil Loss (Mg ha$^{-1}$ yr$^{-1}$) | Runoff (%) | | |
| Trenching + Vegetative barrier | - | 3.4 | - | 27.6 | Black, Hot semi- arid | Khola, 2004 |
| | - | 10.5 | - | 48.5 | Black, Hot semi- arid | Khola, 2004 |
| | - | 7.6 | - | 45.5 | | Khola, 2004 |
| | 0.84 | 10.2 | 1.53 | 21.6 | Alluvial, Hot sub humid | Rao & Reddy, 1996 |
| | 1.18 | 15.5 | 3.26 | 29.3 | | Rao & Reddy, 1996 |
| | 0.93 | 8.7 | 1.55 | 17.6 | Vertisol , Hot semi-arid | Ali et al., 2014 |
| | 0.66 | 4.1 | 1.55 | 17.6 | | Ali et al., 2014 |
| | 0.05 | 0.4 | 1.55 | 17.6 | | Ali et al., 2014 |
| | 6.4 | 10.8 | 19 | 17 | Red laterite, hot sub humid | Sahoo & Adhikari, 2014 |
| | 14 | 12.7 | 23.9 | 17 | | Sahoo & Adhikari, 2014 |
| | 9.9 | 13.4 | 19 | 23.5 | | Sahoo & Adhikari, 2014 |
| | 11 | 15.3 | 23.9 | 23.5 | Red laterite, hot sub humid | Sahoo & Adhikari, 2014 |






**Table 1b.** Details of the experiments and sources of data used to assess relative merits of different
contour grass barriers (CGBs)

<2% slope

| Contour grass barriers | Soil Loss (t ha$^{-1}$) | Runoff (%) | Yield (Kg ha$^{-1}$) | Crop | Soil type and climate | Source |
|---|---|---|---|---|---|---|
| *Cynodon dactylon* | 3.01 | 18 | 1036 | Rice | Red Laterite, Hot | Subudhi and Senapati. 1996 |
| | 5.51 | 16.83 | 1748 | Rice | sub-humid | Subudhi et al. 1998 |
| | 4.73 | 15.59 | 1759 | Rice | Red Laterite, hot | Subudhi et al. 1998 |
| | 4.81 | 15.67 | 1519 | Rice | sub-humid | Subudhi et al. 1998 |
| *Pennisetum perpureum* | 2.68 | 17.4 | 1669 | Rice | Red Laterite, Hot | Subudhi and Senapati. 1996 |
| | 3.05 | 18.1 | 1562 | Rice | sub-humid | Subudhi and Senapati. 1996 |
| | 4.4 | 15.32 | 1828 | Rice | Red Laterite, Hot | Subudhi et al. 1998 |
| | 4.42 | 15.01 | 1925 | Rice | sub-humid | Subudhi et al. 1998 |
| | 4.41 | 15.17 | 1877 | rice | | Subudhi et al. 1998 |
| *Vetiveria zizanoides* | 2.22 | 16.6 | 2133 | Rice | Red Laterite, Hot | Sbudhi and Senapati. 1996 |
| | 4.23 | 14.83 | 2042 | Rice | sub-humid | Subudhi et al. 1998 |
| | 4.02 | 14.05 | 1976 | Rice | Red Laterite, Hot | Subudhi et al. 1998 |
| | 3.96 | 13.88 | 2214 | Rice | sub-humid | Subudhi et al. 1998 |
| | 7.1 | 34.63 | 2000 | Maize | Red Laterite, Hot | Senapati and Sharma. 2007 |
| | 6.89 | 31.59 | 2022 | Maize | sub-humid | Senapati and Sharma. 2007 |
| | 6.48 | 28.31 | 2053 | Maize | | Senapati and Sharma. 2007 |
| | 1.14 | 16.2 | 1377 | Sorghum | Red Laterite, Hot | Prasad et al. 2005 |
| | 0.73 | 13.6 | 699 | Sorghum | sub-humid | Prasad et al. 2005 |
| *Eulaliopsis binnata* | 2.37 | 17.5 | 1436 | Rice | Red Laterite, Hot | Subudhi and Senapati. 1996 |
| | 4.82 | 15.87 | 1933 | Rice | sub-humid | Subudhi et al. 1998 |
| | 5.5 | 16.32 | 1812 | Rice | | Subudhi et al. 1998 |
| | 5.54 | 16.2 | 1769 | rice | | Subudhi et al. 1998 |
| *Cymbopogon martinii* | 2.57 | 17.7 | 1911 | rice | Red Laterite, Hot sub-humid | Subudhi and Senapati. 1996 |
| *Dicanthium annulatum* | 1.05 | 15.5 | 1364 | Sorghum | Black soil, hot-semiarid | Prasad et al. 2005 |
| | 0.69 | 13.7 | 697 | Sorghum | | Prasad et al. 2005 |
| | 0.18 | 7.1 | 808 | Sunflower | | Bhanavase et al. 2007 |
| | 0.85 | 40 | - | | Inceptisol, Black | Bhanavase et al. 2007 |
| | 0.26 | 12.2 | - | | sub-humid | Bhanavase et al. 2007 |
| | 0.3 | 12.5 | - | | | Bhanavase et al. 2007 |
| | 0.52 | 28.14 | - | | | Bhanavase et al. 2007 |
| | | | | | Inceptisol, Black sub-humid | |
| *Cenchrus ciliaris* | 0.14 | 6.5 | 867 | Sunflower | Inceptisol, Black | Bhanavase et al. 2007 |
| | 0.74 | 34.8 | - | | sub-humid | Bhanavase et al. 2007 |
| | 0.21 | 11.6 | - | | Inceptisol, Black | Bhanavase et al. 2007 |
| | 0.22 | 11 | - | | sub-humid | Bhanavase et al. 2007 |
| | 0.46 | 24.12 | - | | | Bhanavase et al. 2007 |
| | 1.01 | 15.8 | | Sorghum | Black soil, hot | Prasad et al. 2005 |
| | 0.77 | 13.9 | 1359 | Sorghum | semi-arid | Prasad et al. 2005 |
| | | | 697 | | | |



| | | | | | | |
|---|---|---|---|---|---|---|
| *Saccharum munja* | 0.86 | 16.3 | 1355 | Sorghum | Black soil, hot | Prasad et al. 2005 |
| | 0.7 | 13.4 | 674 | sorghum | semi-arid | Prasad et al. 2005 |
| *Stylosanthes hemata* | 8.92 | 33.52 | 1789 | Maize | Red laterite, Hot | Senapati and Sharma. 2007 |
| | 8.21 | 33.21 | 1766 | Maize | sub-humid | Senapati and Sharma. 2007 |
| | 8.13 | 34.41 | 1733 | Maize | | Senapati and Sharma. 2007 |
| | 5.81 | 16.87 | 1777 | Rice | | Subudhi et al. 1998 |
| | 5.85 | 16.92 | 1775 | Rice | Red laterite, Hot | Subudhi et al. 1998 |
| | 5.61 | 16.63 | 1803 | Rice | sub-humid | Subudhi et al. 1998 |
| | 2.8 | 18.2 | 1280 | rice | | Sbudhi and Senapati. 1996 |
| *Pennisetum pedicellatum* | 8.01 | 34.01 | 2011 | Maize | Red laterite, Hot | Senapati and Sharma. 2007 |
| | 7.01 | 30.98 | 1990 | Maize | sub-humid | Senapati and Sharma. 2007 |
| | 6.97 | 31.64 | 1969 | Maize | | Senapati and Sharma. 2007 |
| Cultivated fallow | 7.87 | 23.5 | | rice | Red laterite, Hot sub-humid | Sbudhi and Senapati. 1996 |
| Control | 3.47 | 21.4 | 1236 | Rice | Red laterite, Hot | Sbudhi and Senapati. 1996 |
| | 10.39 | 19.94 | 1332 | Rice | sub-humid | Subudhi et al. 1998 |
| | 7.54 | 19.02 | 1330 | Rice | Red laterite, Hot | Subudhi et al. 1998 |
| | 7.24 | 19.18 | 1508 | Rice | sub-humid | Subudhi et al. 1998 |
| | 8.45 | 40.18 | 1720 | Maize | | Senapati and Sharma. 2007 |
| | 9.22 | 42.32 | 1790 | Maize | Red laterite, Hot | Senapati and Sharma. 2007 |
| | 9.02 | 42.6 | 1717 | Maize | sub-humid | Senapati and Sharma. 2007 |
| | 1.89 | 22 | 1140 | Sorghum | Black Soil, hot sub- | Prasad et al. 2005 |
| | 1.45 | 20.2 | 562 | Sorghum | humid | Prasad et al. 2005 |
| | 0.22 | 10.12 | 618 | Sunflower | Inceptisol, Black | Bhanavase et al. 2007 |
| | 1.15 | 53 | - | | sub-humid | Bhanavase et al. 2007 |
| | 0.4 | 15.2 | - | | Inceptisol, Black | Bhanavase et al. 2007 |
| | 0.5 | 16.2 | - | | sub-humid | Bhanavase et al. 2007 |
| | 0.8 | 40.2 | - | | | Bhanavase et al. 2007 |

2-4% slope

| | | | | | | |
|---|---|---|---|---|---|---|
| *Vetiveria zizanoides* | 2.54 | 16.27 | 1075 | Blackgram | Red laterite, Hot | Mishra and Sahu. 2001 |
| | 1.78 | 18.45 | 803 | Blackgram | sub-humid | Mishra and Sahu. 2001 |
| | 3.5 | 27.4 | 5.9 | Sorghum | Alluvial Soil, sub- | Chand and Bhan. 2000 |
| | 7.2 | 33 | 1900 | Maize | humid | Bhardwaj and Sindhwal. 2007 |
| | 9.8 | 43 | 2389 | Maize | Alluvial, Sub-tropical | Bhardwaj and Sindhwal. 2007 |
| | 8.6 | 42 | 2063 | Maize | Alluvial, Sub-tropical | Bhardwaj and Sindhwal. 2007 |
| | 6.9 | 40 | 2042 | Maize | Alluvial, Sub-tropical | Bhardwaj and Sindhwal. 2007 |
| | 2.9 | 22 | 3124 | Maize | Alluvial, Sub-tropical | Bhardwaj and Sindhwal. 2007 |
| | 5 | 30 | 3144 | Maize | | Bhardwaj and Sindhwal. 2007 |
| | 5.5 | 27 | 2278 | Maize | | Bhardwaj and Sindhwal. 2007 |
| | 6.72 | 35.1 | 2444 | Maize | | Bhardwaj and Sindhwal. 2007 |
| | | | | | | Bhardwaj and Sindhwal. 2007 |
| | | | | | | Bhardwaj and Sindhwal. 2007 |
| | | | | | | Bhardwaj and Sindhwal. 2007 |
| *Pennisetum perpureum* | 3.08 | 16.5 | 1002 | Blackgram | Red laterite, Hot | Mishra and Sahu. 2001 |
| | 2.96 | 18.88 | 624 | Blackgram | sub-humid | Mishra and Sahu. 2001 |



| Species | | | | Crop | Soil/Climate | Reference |
|---|---|---|---|---|---|---|
| *Eulaliopsis binata* | 3.15 | 18.24 | 836 | Blackgram | Red laterite, Hot sub-humid | Mishra and Sahu. 2001 |
| | 2.75 | 20.51 | 618 | Blackgram | | Mishra and Sahu. 2001 |
| | 7.9 | 34 | 1869 | Maize | Alluvial, Sub-tropical | Bhardwaj and Sindhwal. 2007 |
| | 10.6 | 46 | 2333 | Maize | | Bhardwaj and Sindhwal. 2007 |
| | 12.4 | 49 | 1833 | Maize | Alluvial, Sub-tropical | |
| | 8.3 | 42 | 1961 | Maize | | Bhardwaj and Sindhwal. 2007 |
| | 3.6 | 25 | 2941 | Maize | Alluvial, Sub-tropical | |
| | 7.3 | 31 | 2839 | Maize | | Bhardwaj and Sindhwal. 2007 |
| | 7.3 | 32 | 2028 | Maize | Alluvial, Sub-tropical | |
| | 8.34 | 37.9 | 2296 | Maize | | Bhardwaj and Sindhwal. 2007 |
| | | | | | | Bhardwaj and Sindhwal. 2007 |
| | | | | | | Bhardwaj and Sindhwal. 2007 |
| | | | | | | Bhardwaj and Sindhwal. 2007 |
| | | | | | | Bhardwaj and Sindhwal. 2007 |
| *Heteropogon contortus* | 0.08 | 5.5 | 523 | Sorghum | Red soil, hot sub-humid | Narayan et al. 2014 |
| | 0.6 | 15.9 | - | | | Narayan et al. 2014 |
| | 0.2 | 4.1 | - | | | Narayan et al. 2014 |
| *Cenchrus ciliaris* | 0.9 | 8.37 | 509 | Sorghum | Red soil, hot sub-humid Red soil, hot sub- humid | Narayan et al. 2014 |
| | 0.82 | 19.4 | - | | | Narayan et al. 2014 |
| | 0.3 | 6.84 | - | | | Narayan et al. 2014 |
| | 4 | 30.2 | 7.2 | Sorghum | Alluvial Soil, sub-humid | Chand and Bhan. 2000 |
| *Pannicum antidotale* | 6.12 | 33.3 | 2460 | Maize | Alluvial, Sub-tropical | Bhardwaj and Sindhwal. 2007 |
| | 5.8 | 29 | 1911 | Maize | | Bhardwaj and Sindhwal. 2007 |
| | 8.1 | 41 | 2528 | Maize | | |
| | 7.6 | 38 | 2073 | Maize | Alluvial, Sub-tropical | Bhardwaj and Sindhwal. 2007 |
| | 6.2 | 39 | 2059 | Maize | | Bhardwaj and Sindhwal. 2007 |
| | 2.9 | 23 | 3109 | Maize | | Bhardwaj and Sindhwal. 2007 |
| | 6.1 | 31 | 3089 | Maize | | Bhardwaj and Sindhwal. 2007 |
| | 6.8 | 28 | 2138 | Maize | | Bhardwaj and Sindhwal. 2007 |
| | | | | | | Bhardwaj and Sindhwal. 2007 |
| | | | | | | Bhardwaj and Sindhwal. 2007 |
| *Saccharum munja* | 3.87 | 18.93 | 963 | Blackgram | Red laterite, Hot sub-humid | Mishra and Sahu. 2001 |
| | 3.07 | 21.04 | 603 | Blackgram | | Mishra and Sahu. 2001 |
| Control | 3.42 | 17.35 | 965 | Blackgram | Red laterite, Hot sub-humid | Mishra and Sahu. 2001 |
| | 3.27 | 20.75 | 603 | Blackgram | | Mishra and Sahu. 2001 |
| | 7.5 | 46.5 | 5.3 | Sorghum | Alluvial Soil, sub-humid | Chand and Bhan. 2000 |
| | 46.28 | 18.04 | - | Maize | | Bhardwaj and Sindhwal. 2007 |
| | 0.41 | 19.8 | 480 | Sorghum | Alluvial, Sub-tropical | Narayan et al. 2014 |
| | 1.4 | 29.7 | - | | | |
| | 1 | 13.7 | - | | Red soil, hot sub- | |





| | | | humid | | |
|---|---|---|---|---|---|
| **>4% slope** | | | | | |
| *Thysanolaena maxima* | 6.92 | 13.85 | 891 | Finger millet | Red laterite, Hot sub-humid | Sudhishri et al. 2008 |
| | 6.02 | 13 | 1105 | | | |
| | 7.16 | 14.06 | 1045 | | | |
| *Vetiver zizanioides* | 4.22 | 8.79 | 1092 | Finger millet | Red laterite, Hot sub-humid | Sudhishri et al. 2008 |
| | 3.85 | 7.85 | 1226 | | | |
| | 4.06 | 9.88 | 1346 | | | |
| | 9.87 | 40.52 | 2180 | | | |
| *Saccharum munja* | 4.49 | 9.36 | 1045 | Finger millet | Red laterite, Hot sub-humid | Sudhishri et al. 2008 |
| | 4.02 | 8.25 | 1226 | | | |
| | 4.65 | 10.83 | 1427 | | | |
| *Cynodon dactylon* | 2.1 | 27.1 | 4355 | | Alluvial (Entisols) sub humid tropical | Narain et al. 1994 |
| *Dicanthium annulatum* | 1.02 | 21.2 | 6805 | | Alluvial (Entisols) sub humid tropical | Narain et al. 1994 |
| | 0.23 | 1.9 | | | | |
| *Eulaliopsis binnata* | 0.29 | 5.2 | 16290 | | Alluvial (Entisols) sub humid tropical | Narain et al. 1994 |
| *Chrysopogon fulvus* | 0.3 | 2.5 | 19170 | | Alluvial (Entisols) sub humid tropical | Narain et al. 1994 |
| Control | 83.04 | 32.6 | - | | Alluvial (Entisols) sub humid tropical | Narain et al. 1994 |
| | 18.45 | 16.2 | - | | | Narain et al. 1994 |
| | 92.42 | 71.1 | - | | | Narain et al. 1994 |
| | 13.9 | 26.02 | 607 | Finger millet | Red laterite, Hot sub-humid | Sudhishri et al. 2008 |
| | 13.7 | 24.84 | 676 | | | Sudhishri et al. 2008 |
| | 14.28 | 26.78 | 682 | | | Sudhishri et al. 2008 |









**Table 2.** Various attributes and normalized scores used for calculating CPI for different vegetative barriers

| | Vetiver | Hy. Napier | Panicum | Dicanthium | Broom | Cymbopogon | congosignal | Eulaliopsis | Saccharum munja Sambuta |
|---|---|---|---|---|---|---|---|---|---|
| IR (cm ha⁻¹) | 7.5-11.6 | 11.08-13.26 | 7.68-8.21 | 7.12-8.02 | 14.42 | | 20.06 | 7.7-8.6 | 12.7 |
| Score | 1.0 | 1.0 | 1.0 | 1.0 | 1.0 | 1.0 | 1.0 | 1.0 | 1.0 |
| Soil Loss (t ha⁻¹) | 1.0-9.8 | 2.96-3.08 | 2.9-8.1 | 0.2-1.02 | 6.02-7.16 | 1-2.57 | 6.0-8.0 | 2.75-12.4 | 4.02-4.7 |
| Score | 0.5 | 0.8 | 0.5 | 1.0 | 0.5 | 1.0 | 0.5 | 0.8 | 0.8 |
| Soil binding (ml mm⁻²) | 206-248 | 577-803 | 82-127 | 127-331 | 153-178 | 80-150 | 75-110 | 613-956 | 230-395 |
| Score | 0.2 | 1.0 | 0.2 | 0.2 | 0.2 | 0.2 | 0.2 | 1.0 | 0.5 |
| Sod forming Soil depth (cm) | 60-90 | 40-60 | 80-105 | 50-60 | 60-95 | 80-100 | 30-50 | 30-40 | 35-50 |
| Score | 0.8 | 0.5 | 1.0 | 0.5 | 0.8 | 1.0 | 0.5 | 0.2 | 0.2 |
| Fodder or commercial value | Average | excellent | Very good | Excellent | Average | Average | Good | Very good | Average |
| Score | 0.2 | 1.0 | 0.5 | 1.0 | 0.2 | 0.2 | 0.5 | 1.0 | 0.2 |
| Cost of establishme-nt (Rs) | 3500-4500 | 4000-6000 | 3000-5000 | 3500-4500 | 4000-5000 | 4000-6000 | 2500-3500 | 3500-5000 | 3000-4501 |
| Score | 0.5 | 0.5 | 0.5 | 0.5 | 0.5 | 0.5 | 0.8 | 0.5 | 0.8 |





**Table 3.** Impact of grasses in arresting Soil loss and Runoff

| Treatment | Run off (%) | Soil loss (Mg ha$^{-1}$ yr$^{-1}$) | Number of Samples (n) |
|---|---|---|---|
| **Vegetative barrier** | | | |
| Control (Without Grass) | 11.26 – 62.40 (37.71 ± 18.12)$^{a}$ | 0.53-30.90 (9.56±8.79)$^{a}$ | 25 |
| With Grass | 5.87 – 44.10 (20.93 ± 10.76)$^{b}$ | 0.5-18.7 (3.93±5.03)$^{b}$ | 25 |
| Conservation use efficiency (CUE) | 44.56 | 59.04 | 25 |
| **Along with Structural Conservation measures** | | | |
| Control (Grazed) | 17.0 – 48.5 (25.53 ± 10.88)$^{a}$ | 1.53-3.26 (1.88±0.77)$^{a}$ | 17 |
| Combination | 0.40 – 15.30 (9.37± 4.76)$^{a}$ | 0.05-1.18 (0.73±0.42)$^{a}$ | 17 |
| Conservation use efficiency (CUE) | 62.93 | 60.96 | 17 |
| **Grazing Management** | | | |
| Control (Grazed) | 11.30 – 33.4 (24.33 ± 6.55)$^{a}$ | 1.52-3.28 (2.58±0.73)$^{a}$ | 12 |
| Management | 6.60 – 22.2 (14.12 ± 5.23)$^{b}$ | 0.58-1.3 (0.95±0.29)$^{b}$ | 12 |
| Conservation use efficiency (CUE) | 42.01 | 63.18 | 12 |

Values in the parentheses are mean ± SD

Different letters in the same column are significantly different at P<0.05






**Table 4.** Site specific suitable grasses for Contour Vegetative Barriers

| S.No. | State | Crop | Barrier |
|---|---|---|---|
| 1 | Andhra Pradesh | Sorghum/castor | *Cenchrus ciliaris (* Buffel grass) |
| 2 | Haryana | Urd<br>Bajra and Wheat | Mixed barrier of *Vetiveria zizaniodes* (Vetiver) plus *Eulaliopsis binata* (Sabai grass) |
| 3 | Karnataka | Groundnut<br>Finger millet<br>Sorghum | *Vetiveria zizaniodes* (Vetiver) on contour<br>Combination of graded bund and *Vetiveria zizaniodes* (Vetiver)<br>Compartmental bunding with *Vetiveria zizanoides* (Vetiver) |
| 4 | Madhya Pradesh | Soyabean | *Cympogon martini* (Lemon grass/ Palmarosa) |
| 5 | Maharashtra | Sorghum, Cotton | *Vetiveria zizanoides* (Vetiver) |
| 6 | Orissa | Paddy<br>Cowpea (green pond) | *Vetiveria zizanoides* (Vetiver) , *Cynodon dactylon* (Bermuda grass) |
| 7 | Punjab | Maize | *Saccharum sps.* |
| 8 | Tamil Nadu | Potato | *Pennisetum purpureum* (Napier/Elephant grass) |
| 9 | Uttarakhand | Corn | *Panicum maximum* (Guinea/buffalo grass) |




















**Table 5.** Relative merits of contour grass barrier (CGBs) in different land slopes

| Treatment | Run off (%) | Soil loss (Mg ha⁻¹ yr⁻¹) | Yield | Number of Samples (n) |
|---|---|---|---|---|
| <2 % slope | | | | |
| Control (Without Grass) | 10.12 – 42.60 (27.10 ± 13.58) [a] | 0.22 – 10.39 (5.03 ± 3.92) [a] | 546 – 1717 (1179 ± 475.32) [a] | 12 |
| With Grass | 13.88 – 16.92 (15.81 ± 1.06) [b] | 3.82 – 5.85 (5.03 ± 0.69) [a] | 1519 – 2214 (1843 ± 176.09) [b] | 12 |
| Relative reversibility | 52.64 % (Δ RF) | Insignificant (Δ SL) | 44 %( Δ Y) | |
| 2-4 % slope | | | | |
| Control (Without Grass) | 13.20 – 71.10 (28.36 ± 15.36) [a] | 0.41 – 92.42 (23.46 ± 32.54) [a] | 345 – 965 (756 ± 341.17) [a] | 12 |
| With Grass | 16.27 – 41.00 (24.65 ± 9.45) [a] | 1.78 – 8.10 (4.24 ± 2.11) [b] | 618 – 2528 (1257 ± 684.69) [b] | 12 |
| Relative reversibility | 14.63 % (Δ RF) | 141 % (Δ SL) | 53 %( Δ Y) | |
| > 4 % slope | | | | |
| Control (Without Grass) | 24.84 – 71.10 (36.27 ± 19.70) [a] | 13.70 – 92.42 (43.47 ± 40.53) [a] | 558-682 (638 ± 53.80) [a] | 5 |
| With Grass | 7.85 – 14.06 (11.51 ± 3.39) [b] | 3.85 – 7.16 (5.63 ± 1.52) [b] | 891 – 1226 (1071 ± 121.13) [b] | 5 |
| Relative reversibility | 55.54 % (Δ RF) | 107 % (Δ SL) | 50.64 %( Δ Y) | |



Relative reversibility of erosion/Water loss-
$\Delta$ Erosion/ $\Delta$ Runoff = $\dfrac{\text{Erosion/ water loss without CGB} - \text{Erosion/ water loss with CGB}}{\text{Mean erosion/ water loss}}$ X100

Relative yield gain -
$\Delta$ Yield gain = $\dfrac{\text{Mean yield with CGB} - \text{Mean yield without CGB}}{\text{Mean yield}}$ X 100






**Table 6.** Comparative Comprehensive Performance Index of vegetative barrier

| Wt. | Vetiver | Hy. Napier | Panicum | Dicanthium | Broom | Cymbopogon | Congosignal | Eulaliopsis | Sambuta |
|---|---|---|---|---|---|---|---|---|---|
| IR (0.2) | 1.0 | 1.0 | 1.0 | 1.0 | 1.0 | 1.0 | 1.0 | 1.0 | 1.0 |
| Wt x score | 0.2 | 0.2 | 0.2 | 0.2 | 0.2 | 0.2 | 0.2 | 0.2 | 0.2 |
| Soil loss (0.2) | 0.5 | 0.8 | 0.5 | 1.0 | 0.5 | 1.0 | 0.5 | 0.8 | 0.8 |
| Wt x score | 0.10 | 0.16 | 0.10 | 0.2 | 0.1 | 0.2 | 0.1 | 0.16 | 0.16 |
| Soil binding (0.1) | 0.2 | 1.0 | 0.2 | 0.2 | 0.2 | 0.2 | 0.2 | 1.0 | 0.5 |
| Wt x score | 0.02 | 0.1 | 0.02 | 0.02 | 0.02 | 0.02 | 0.02 | 0.1 | 0.05 |
| Sod forming (0.1) | 0.8 | 0.5 | 1.0 | 0.5 | 0.8 | 1.0 | 0.5 | 0.2 | 0.2 |
| Wt x score | 0.08 | 0.05 | 0.1 | 0.05 | 0.08 | 0.1 | 0.05 | 0.02 | 0.02 |
| Fodder value (0.2) | 0.2 | 1.0 | 0.5 | 1.0 | 0.8 | 0.2 | 0.5 | 0.5 | 1.0 |
| Wt x score | 0.04 | 0.2 | 0.1 | 0.2 | 0.16 | 0.04 | 0.1 | 0.1 | 0.2 |
| Cost established (0.2) | 0.5 | 0.5 | 0.5 | 0.5 | 0.8 | 0.5 | 0.8 | 0.5 | 0.8 |
| Wt x score | 0.1 | 0.1 | 0.1 | 0.1 | 0.16 | 0.1 | 0.16 | 0.1 | 0.16 |
| CPI=Σ (Wt x score) | 0.54 | 0.81 | 0.62 | 0.77 | 0.72 | 0.63 | 0.68 | 0.68 | 0.79 |

Values in the parentheses are weights assigned to the respective attributes





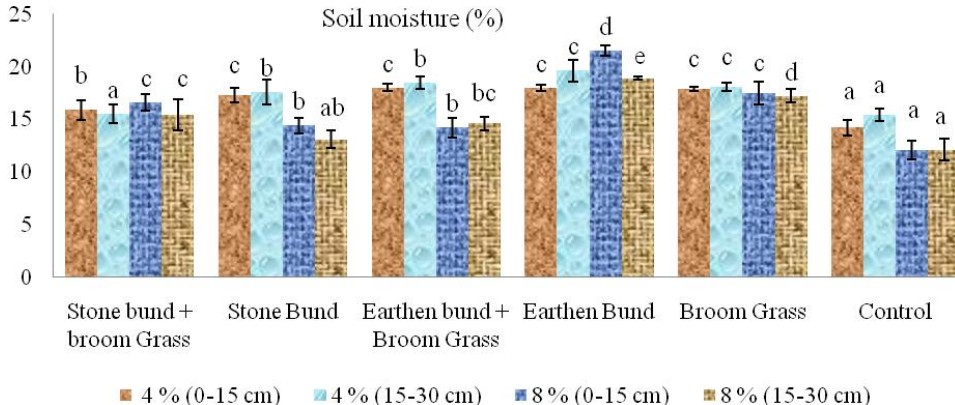


**Figure 1**. Complimentary role of grasses in enhancing soil profile moisture at 4 % and 8 % slope. Values with different letters are significantly different at 95% confidence level (p ≤ 0.05; ANOVA-DMRT).

















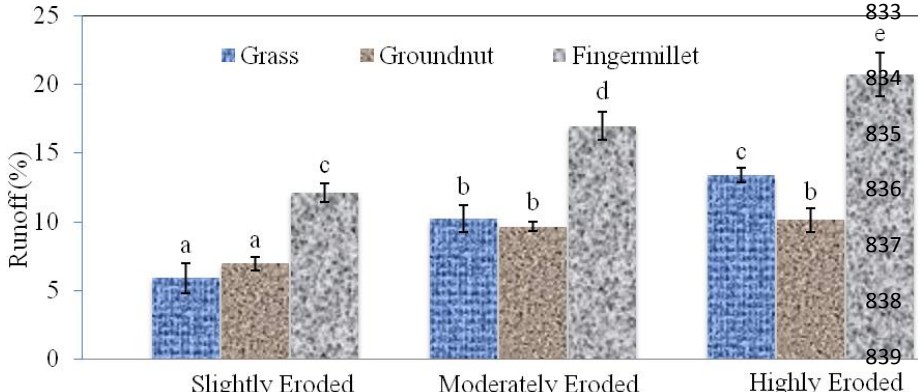










**Figure 2**. Impact of grasses in reducing runoff in lateritic soil. Values with different letters are significantly different at 95% confidence level (p ≤ 0.05; ANOVA-DMRT).
























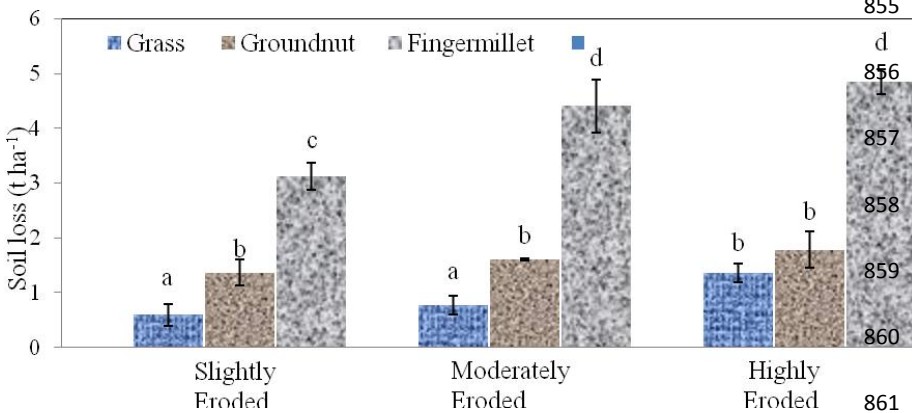


**Figure 3**. Impact of grasses in reducing soil loss in lateritic soil. Values with different letters are significantly
different at 95% confidence level (p ≤ 0.05; ANOVA-DMRT).














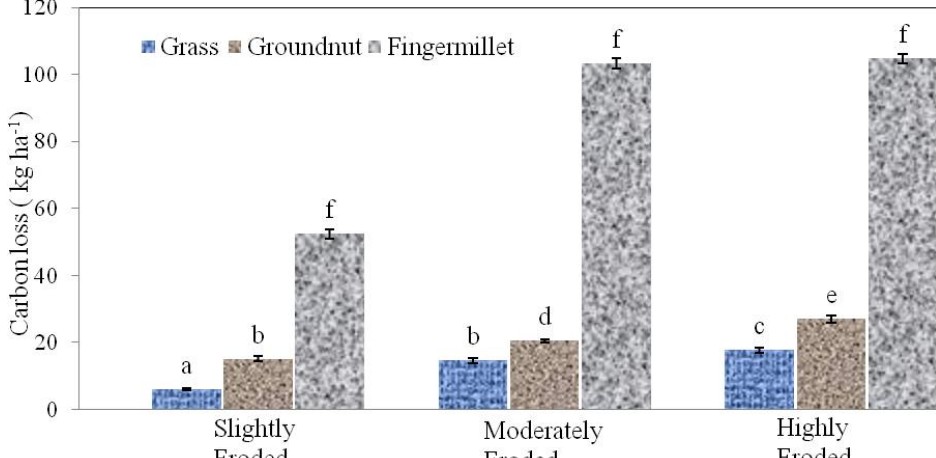


**Figure 4**. Impact of grasses in reducing carbon loss in lateritic soil. Values with different letters are significantly different at 95% confidence level (p ≤ 0.05; ANOVA-DMRT).

















**List of Tables**




