# Peer review of "Reversing land degradation through grasses: a systematic meta-analysis in the Indian tropics"

_Solid Earth, 2016_

## Referee Comment (RC1) · Anonymous Referee #1 · 17 Nov 2016

The manuscript se-2016-143 is an interesting exploration of reversing land degradation through re vegetation of suitable grasses and is interesting to the wide readership of Solid Earth. In fact the topic is pertinent and addresses a challenging sustainability concern. Nevertheless, I have some suggestions for further improving the clarity and organization of the manuscript.

The abstract should be rewritten and it must start with some background information of the problem. Moreover, it must end up with a strong take home message.

Similarly, the criteria used for selecting various papers for this particular meta-analysis must be clearly provided in Materials and Methods sections.

---

## Referee Comment (RC2) · Anonymous Referee #2 · 26 Nov 2016

Reversing land degradation through grasses: a systematic meta-analysis in the Indian tropics Solid Earth Discuss., doi:10.5194/se-2016-143, 2016 Comments to the author Land degradation through desertification produces many negative 'on-site' and 'off-site' effects, especially during intense seasonal rainfall events. Often substantive topsoil removal occurs by flowing water, leading to decreased soil quality, nutrient loss and reduced infiltration, which produces a positive feedback by increasing run-off and hence further accelerating soil erosion. The present manuscript deals with the topic which critically analyzes the effect of grasses in reversing the process of land degradation. Grass species have tremendous potentialities in soil conservation as it has a great binding influence on soil particles. The authors clearly spell out the advantage

of vegetative barriers of Pennisetum purpureum and Saccharum munja in controlling soil loss due to water erosion and also it checks the loss of soil carbon due to erosion. The manuscript revealed that by managing the grassland with cut & carry system, rotational grazing and control grazing can greatly reduce the water and soil loss and helps in the reversing the land degradation process. Few points which need attention are mentioned below. 1. The abstract should start with the introductory remark and should end with the concluding remark. 2. There are few typographical errors which needs correction. Authors are requested to check the entire manuscript for typographical error. 3. Data on soil moisture storage, erosion aspect and carbon sequestration/build up may be strengthened in result and discussion section. Comments to the editor

In my opinion, the manuscript addresses a topic which should be of general interest to readers of Solid Earth Discussions. As it stands the interpretation is basic and the way the information is presented is good. The paper generates some useful information on efficacy of vegetative barriers for controlling/reversing land degradation. The manuscript is well written. Looking into the quantum and quality of data, the manuscript should be published in Journal of Solid Earth Discussions.

---

## Author Comment (AC1) · 17 Dec 2016

Dear Editor, Thank you very much for completing the review evaluation and providing valuable suggestions. We have addressed all the comments clearly and revised the manuscript accordingly.

Comments from Referees: The abstract should be rewritten and it must start with some background information of the problem. Moreover, it must end up with a strong take home message.

Author's response: The abstract is rewritten as suggested. We started the abstract with the following sentences.

[Figure]

Although intensive agriculture is necessary to sustain the world's growing population, accelerated soil erosion contribute to a decrease in the environmental health of ecosystems at local, regional and global scales. Reversing the process of land degradation using vegetative measures is utmost important in such lands.

Similarly, at the end of the abstract we have given some take home messages that has been emerged out from the study.

The present analysis also indicated that grass must be used as vegetative strip to maintain soil quality in sloppy arable areas (8.5 m ha) of Indian hilly regions. Similarly, due attention should be given for establishing grasses to about 3 m ha degraded pasture lands and 3.5 m ha shifting cultivation areas in India to reverse the land degradation.

Comments from Referees: The criteria used for selecting various papers for this particular meta-analysis must be clearly provided in Materials and Methods sections.

Author's response: The criteria used for selecting various papers for this particular meta-analysis is given in the materials and methods section where the authors clearly stated the basis of selecting the various papers for all three roles of grasses.

———————————————

---

## Author Comment (AC2) · 17 Dec 2016

Dear Editor, Thank you very much for completing the review evaluation and providing valuable suggestions. We have addressed all the comments clearly and revised the manuscript accordingly.

Comments from Referees: The abstract should start with the introductory remark and should end with the concluding remark.

Author's response: The abstract has been revised as suggested by the Referee 1. In fact this suggestion has been given by both the Referees.

Comments from Referees: There are few typographical errors which needs correction.

[Figure]

Authors are requested to check the entire manuscript for typographical error.

Author's response: The manuscript has been thoroughly checked by the authors. Obvious typographical errors have been corrected.

Comments from Referees: Data on soil moisture storage, erosion aspect and carbon sequestration/build up may be strengthened in result and discussion section.

Author's response: The criteria for selecting the evidences on the role of grasses on erosion has already highlighted in the materials and methods. The data on soil moisture storage and carbon sequestration are very limited. Available information on these issues is given.